# Transnational Child Sexual Abuse: Outcomes from a Roundtable Discussion

**DOI:** 10.3390/ijerph16020243

**Published:** 2019-01-16

**Authors:** Hannah L. Merdian, Derek E. Perkins, Stephen D. Webster, Darragh McCashin

**Affiliations:** 1School of Psychology, University of Lincoln, Lincoln LN6 7TS, UK; 2School of Law, Royal Holloway University of London, Egham, Surrey TW20 OEX, UK; Derek.Perkins@rhul.ac.uk; 3Independent Research Psychologist, London SM6 9AA, UK; stephenwebster@pivotalresearchservices.com; 4School of Psychology, University College Dublin, Belfield, Dublin 4, Ireland; mccashid@tcd.ie

**Keywords:** child sexual abuse, child sexual exploitation, transnational child sexual abuse, travelling offender

## Abstract

The phenomenon of men who travel across international borders to engage in child sexual abuse presents significant public health, legal, diplomatic, cultural, and research challenges. Briefed in the current scope of this issue by relevant stakeholders across legislation, research, and social policy, a roundtable discussion held in London aimed to synthesise plenary discussions from multidisciplinary attendees across law enforcement, academia, non-profit, and industry sectors with direct relevance to the UK. Specifically, the roundtable discussions aimed to gather the central themes relating to attendee discussions on the key challenges, affected countries, response strategies, and knowledge gaps. Four key themes were identified from the data, relating to the definition of Transnational Child Sexual Abuse (TCSA), criminal justice, geographical considerations, and issues surrounding tourism/hospitality. The data highlighted four priorities for future development and research, namely developing offender typologies, victim-centric investigative practice, prevalence and definitions, and collaborations. These themes provide insight into the issue of transnational child sexual abuse from the perspective of different disciplines and offer a strategy to prioritise, and collaborate, in the efforts against transnational child sexual abuse.

## 1. Introduction

The role of offenders travelling within and across national borders to plan, prepare, and carry out sexual offences against children is a significant public protection concern [1,2]. Transnational Child Sexual Abuse (TCSA) can refer to: (*a*) offenders, including those with a criminal history, travelling to a different jurisdiction and engaging in the sexual abuse of children; (*b*) offenders who intentionally reside abroad for offending purposes (semi-permanent or permanent residence); and (*c*) offenders who engage in Internet-enabled sex offences, utilising technologies such as webcams and live-streaming. It is impossible to be precise about the extent of child sexual abuse as a result of offenders networking and then travelling within the UK and/or internationally due to the hidden and under-reported nature of this offending ([3], para 51). In 2012, the UK Child Exploitation and Online Protection Centre (CEOP) [4] received 1,145 reports of online child sexual exploitation, with the offender looking to travel physically within the UK to meet a child to commit contact sexual abuse. A review of sexual offence prevention orders [5] showed that between 2008 and 2012, 303 British sex offenders had been arrested abroad for child sexual offences. CEOP’s [4] analysis of online chat forums pointed to the dynamic nature of the geographical locations within organised commercial child sexual exploitation, because of increased awareness in destination countries, and offenders moving towards countries that have suffered environmental or political upheaval, leading to displacement of children from their families and a differential focus of local authorities. Given the risk presented by offenders (and groups) networking and travelling within the UK and/or internationally to sexually offend, CEOP listed TCSA, and the role of technology in its conduct, as a key threat in their strategic action plan. 

There is a significant dearth of empirical literature on the issue of TCSA. The majority of published material relates to organised child abuse networks and child trafficking, and focuses on conceptual considerations [6,7,8,9] and policing/legal issues [10,11,12,13,14], resulting in a lack of empirical data on the offenders or the networks they use. However, there is an emerging body of research on the use of cyberspace for the sexual grooming of minors and the adaption of sexual crime theory for online sexual behaviour that can be organised under four broad themes: offender characteristics, types of networks, the role of cyberspace as an enabler and mode of child sexual exploitation, and national and international policy responses. 

### 1.1. Offender Characteristics

The online solicitation of children for sexually-oriented interactions can occur through various methods, with the most prominent being direct engagement with a potential victim through online channels (e.g., chat rooms, instant messaging, email, mobile phones). The National Centre for Missing and Exploited Children [15] survey of online child victimization showed that nearly a quarter of minors who reported having received an unwanted sexual contact online were also asked by the offender to meet in person. Consequently, the research to date has mainly focused on the use and function of online networking in the victim grooming process [16,17,18], offender characteristics [19,20], the role of sexual arousal, emotion regulation, shame, and deception in the online solicitation of a minor [21,22], or the “personas” adopted by offenders in the grooming process [23,24], without a specific focus on transnational child sex offending. While previous research, including Harkins and Dixon [1], identified some behavioural characteristics of child sexual abuse networks, to date, there is no detailed account of the demographic characteristics of offenders, and the role these networks play in the normalization and facilitation of child sexual abuse, given that these provide users with an environment to explore their sexual interests [25,26,27]. Studies investigating organised transnational child sexual exploitation [28,29] have focused on victim identification, victim characteristics, and their engagement rather than on the perpetrators. What is known is that almost all of the offenders reported to CEOP as suspected or convicted transnational child sex abusers were reportedly male (99%), with a median age of above 40 years [4]. 

Based on descriptive analysis of their case information, CEOP suggested a continuum of transnational offending behaviour: *Transient offending* describes the sexual abuse of children, potentially facilitated by the commercial sexual exploitation of children, where the offender will be in short-term contact with the victim(s) (also referred to as “child sex tourism”). *Embedded offending* is the repeated victimisation of a child by the same offender, often accompanied by extended grooming of the victim, significant others, and the community; here, perpetrators may be more likely to reside long-term in the overseas destination. While this describes a topological crime classification, Briggs et al. [30] suggested a motivational typology of online solicitation offenders, based on the analysis of chat logs between offenders and their victims. According to Briggs et al., *fantasy-driven users* tend to maintain their engagement with the minor solely in the online space, whereas *contact-driven users* are focused on transferring the online engagement into a real-life encounter. In keeping with advancing evidence regarding offender typologies, a recent law enforcement perspective document published by *Europol* [31] recommended different strategies for the prevention and management of online sexual coercion and extortion of children to reflect the differences in perpetrators’ motivation and profiles.

### 1.2. Types of Networks

Online communities have established themselves as an important networking and exchange platform for users with a sexual interest in children [32]. There are some paedophile online organisations such as the *North American Man-Boy Love Association*, *Girlchat*, *Boywiki* [16,27,33]; *Cherry-Popping Daddies*, or *Young Petals* [24] who advocate legalising sexual behaviour with minors and the liberalisation of existing child protection laws. These communities play an important role in normalising and validating paedophile intentions, and facilitate the establishment of contacts to other users with deviant sexual interests [34]. As Tate [35] described, many paedophile groups present themselves as suppressed minorities, which serves as a sustainable coherence factor for their participants. These fora often function as a validating source for offence-supportive cognitions [25,26,33,36]. 

While research on online sexual grooming of minors has increased in recent years [17,30], organised child abuse networks remain significantly under-researched. These groups can be defined as a set of offenders who cooperate to recruit children to be abused, pass victims between them, and may be involved in group abuse sessions with children [37]. There is evidence of these rings abusing children offline in contexts such as residential care homes [38], “child sex tourism” in South East Asia [39], as well as delivering abuse in online settings, for example the *Orchid Club* in North America [34]. Burgess [40] distinguished between *transitional* (unorganised) and *syndicated* (highly organised) sex-rings; however, his theory precedes the introduction of the Internet, and thus may not (fully) apply to cyberspace-enabled offending. In a more recent paper on groups that sexually abuse children, Cockbain, Brayley, and Laycock [10] described pathways to group involvement, the perceived benefits derived from group involvement, and the evolution of the group. However, this study is based on analysis of three offenders, limiting the extent to which these findings can be generalised. Generalisation could potentially occur based on the psychological processes supporting and enabling the offending behaviour, for example, the normalizing context of the online group [26,41] and the role of conformity in groups [1]. There has, however, been little work applying these theories to child sexual abuse networks [1,24]. 

### 1.3. The Role of Cyberspace as an Enabler and Mode of Child Sexual Exploitation

A body of work is emerging about the role of cyberspace and the associated behaviours it may encourage as facilitators of child sexual exploitation [1,42,43]. In the context of child sex tourism, Panko and George [8] described websites that covertly or overtly advertise child abuse as a tourism experience. The proliferation of peer-to-peer networks is argued to make the exchange of information easier for offenders and more difficult for law enforcement to intervene [4,24,44]. An emerging risk is the organised online abuse by British nationals of children living abroad [4], but to date no published research on this behaviour was identified. 

A high proportion of online abuse and exploitation crimes involve children, and their own activities concerning peer-related sexual content has meant that young people are also creating potentially illegal materials [45]. Whilst guidelines published by the National Institute for Health and Care Excellence [46] have differentiated between sexually abusive behaviour and those behaviours that are detrimental to a child’s development, the links between harmful sexual behaviours, pornography, grooming, *sexting*, and contact sexual offending are poorly understood and conceptualised [45]. The online behaviours of young people, illustrated by large-scale studies such as *EU Kids Online*, *n* = 18,709 [47], have highlighted the risk of receiving sexual messages; about one quarter of those receiving such messages self-reported that they had experienced negative emotions resulting from them. Whilst this study showed a range of positive outcomes relating to the digital lives of young people, it also highlighted some of the risks linked to online behaviours. For example, 30% of European children (aged 9 to 16 years) reported having communicated with someone they have not met face-to-face, 9% of children have met them in person, and 1% felt negatively affected by it. It is within these data trends that the concerns about transnational sexual offending (abetted by technology) becomes further amplified. 

### 1.4. National and International Policy—Current Response Frameworks

Concern about people networking and travelling within or outside the UK to sexually abuse children has moved higher up the political agenda [2]. In the UK, Sexual Harm Prevention Orders and Sexual Risk Orders were introduced in order to restrict the movement of people who present a risk of harm to children in the UK and abroad [48]. Alongside legislation, the UK Government has prioritised to investigate and prevent child sexual abuse. In order to tackle the issue of British registered sex offenders targeting child-focused organisations abroad, CEOP’s and the Association of Chief Police Officers’ Criminal Records Office launched the International Child Protection Certificate, which allows organisations to check the criminal record of those seeking work with children abroad (https://www.acro.police.uk/icpc/). However, the certificate will not detect offenders without a criminal record in the UK. In addition, there is no mechanism to mandate organisations abroad to use the certificate. On an international level, travel for the purpose of offending behaviour and online grooming have been considered within the Directive 2011/93/EU of the European Parliament and of the Council of 13 December 2011 on combating the sexual abuse and sexual exploitation of children and child pornography [49].

Sexual exploitation of children occurs within more general social, economic, and cultural considerations [50]. Thus it is important to develop comprehensive and pro-active prevention measures that target a number of potential risk factors, such as low levels of education, poverty, ignorance, official corruption, apathy, and a lack of law enforcement and government policy [39,51]. Increased international and local police collaboration has had a positive impact on reducing commercial child sexual exploitation in at-risk countries; for example, Thailand reported reduced crime rates for 2011/2 [4]. In addition, the international non-governmental organisation network ECPAT (Every Child Protected Against Trafficking) has played a significant role in developing measures to reduce the vulnerability of children abroad and has mobilised the tourism industry to take an active part in countering the risk of sexual offending [51]. In 1998, the Code of Conduct for the Protection of Children from Sexual Exploitation in Travel and Tourism (the Child Protection Code) was launched to establish an ethical policy against the commercial sexual exploitation of children, to train tourism personnel, and to provide information to travelers [52]. However, there is very little empirical evidence to inform tourism personnel about the characteristics of networks and offenders therein, how they operate online and offline, and the key touch points of risk in the UK and abroad. Particularly under-researched areas are the social and cultural factors affecting transnational organised child sexual exploitation. The Model National Response (MNR) of the WePROTECT Global Alliance [53] has provided an overview of how nations can establish and deliver a coordinated response to online child sexual abuse and enhance international cooperation, irrespective of the starting point of current policies. Within the MNR, it is recognised that cooperation is needed between law enforcement, the tourism industry, technology sectors, academia, the non-profit sector, charities, global governance bodies, the public, and indeed young people themselves. However, as documented by ECPAT research [54], despite advancing child protection policies, there remains a growth of sexual exploitation of children in travel and tourism across the globe. There are key challenges pertaining to multi-stakeholder engagement at a global level, in addition to resourcing limitations that necessitate prioritisation of law enforcement objectives. Consequently, it is likely that a significant portion of TCSA remains unaddressed. 

## 2. Method

### 2.1. Study Design

Given the limited knowledge surrounding the issues of TCSA, the aim of this roundtable discussion was to collate information from an international, multi-disciplinary, multi-agency perspective that will be of practical value and assistance to those working to combat this particular offending behaviour. The agenda for the roundtable discussion was: (1) to hear an overview of current research findings on transnational child sexual exploitation/abuse, (2) to share an update on current, related transnational law enforcement issues, and (3) to take part in small-group and roundtable scoping and discussion sessions on current issues facing law enforcement, policy makers, offender managers, and treatment providers. The choice of a roundtable discussion format using the procedural details outlined in McCartan et al. [55] was to allow participants to engage in exploratory in-depth interactions with different professionals in order to elicit a rich qualitative dataset.

### 2.2. Participants/Discussants

Thirteen discussants attended the roundtable discussion, representing three sectors: (1) criminal justice and policy-making, (2) child protection/children’s rights organisations, and (3) academia including criminology, psychology, marketing, hospitality/tourism, and childhood development. All participants were directly involved with the topic of online or transnational child sexual abuse in their respective sector; and were either mid-career or senior within their present roles. Participants were recruited via snowball sampling [56], by targeting relevant organisations and lead contributors in the field. In order to facilitate free exploration of this challenging topic, participants were assured of confidentiality by having their identifying organisation removed from the data. 

### 2.3. Procedure and Materials

The roundtable discussion lasted four hours, including breaks, and was facilitated by two independent facilitators from the host institution’s Research and Enterprise Department. Participants were seated at three tables consisting of five attendees and completed an information sheet, highlighting issues of current concern. Following an introductory address from policing and academia lasting approximately 40 minutes each (with a short question and answer period), a first round of small-group discussions followed on the current challenges and issues in addressing TCSA, including affected countries. Using their notes, a representative from each table then communicated back to the full participant group the key points from the discussions at their table. Emerging issues were collected and displayed on large posters across the room, which were then anonymously rated by participants in order to identify the most pressing concerns. Participants used stickers to select the issues they believed to be of highest priority, and assigned them to the respective poster during the interval. These set the agenda for the second part of the group discussions, including the identification of specific strategies to address the identified issues, to deal with at-risk countries, and to highlight further knowledge gaps. For these initial discussions, participants selected the next group discussion that was of most interest to them. 

All discussions were recorded using an encrypted voice-recorder, and additional note-takers were present on each table. All discussions were facilitated to be unstructured in nature to allow participants to explore the topics that were of highest priority to them, based on the developing themes of the event. The independent facilitators acted as a neutral point of contact to mediate any potentially different ideological or power dynamics. 

### 2.4. Data Analysis

All data were initially transcribed. Using an inductive approach, data were analysed using qualitative thematic analysis [57] on the recorded data and notes taken during the discussion. This approach was chosen due to its flexibility and interpretivist epistemology [55]. The posters, post-it notes, and information sheets generated during the course of the day were used to contextualise the primary data from the recordings and notes. Following the familiarisation process of re-reading the data, an initial set of codes was produced by the fourth author. A sample transcript was coded by the first author researcher to ensure inter-rater reliability. Any disagreement or lack of clarity was discussed between researchers until consensus was reached. All codes were then grouped into potential themes and reviewed by the third author. The overall themes were then defined and named. 

## 3. Results

The information sheets showed that the majority of participants were mainly interested in, and advocating for, inter-agency and inter-professional information exchange. Desired outcomes were linked to the development of more insight, new ideas to tackle TCSA in their respective roles, and to make new contacts.

### 3.1. Roundtable Discussion

In the first part of the roundtable discussion, attendees were asked to identify the key challenges and issues of TCSA in their respective discussion groups. Four key themes were identified, concerning issues with the definition of TCSA, criminal justice, geographical considerations, and issues surrounding tourism/hospitality.

#### 3.1.1. TCSA Definition

Participants wondered how TCSA compares to trafficking and online child sexual abuse:
“*I think there’s a more fundamental question to be asked—what do we mean by location and distribution channels? What is being distributed via what channel? (…) It can be online or offline, it can be abroad, or being streamed.*”“*There’s a burning question about motivation there, isn’t there? Because there will be people that will have a sort of prima facie motivation to go to a different country or location—regardless of whether its online/offline—to abuse a child. And so will therefore proactively seek out information that will allow them to do this. But then (…) you have situational offenders who may just be, for example, engaging with CSEM [Child Sexual Exploitation Material] in a network (…) where there’s some on-going conversations, and then all of a sudden, the opportunity presented itself.*”

In this context, participants also highlighted that definitional clarity is needed for *travelling offender* and *victim*. In addition, participants wondered about the wider context of TCSA:
“*[We need to consider] economic discrepancies or poverty in the countries where this happens and the cultural acceptability of the exploitation happening. It’s not just a question of demand—but children coming round hotels offering services … so that parameter of wealth, poverty and social inequality in these locations seemed to be a key factor in our discussions.*”

Following on from these discussions, participants also wondered if there are vulnerable occupations (such as pilots, truck drivers, global event facilitators) that could be targeted for prevention work. 

#### 3.1.2. Criminal Justice Issues

Under this second theme, four sub-themes could be defined:

##### Management of Travelling Offenders

Participants raised the issue of managing travel applications of registered sex offenders, and those who travel without travel application; and who holds responsibility for incidents of sexual abuse where such travelers access countries with no border control or without monitoring systems. Participants also shared knowledge about specific hotspot countries worthy of more focus:
“*If we’re concerned about UK offenders travelling abroad, there’s places like Kenya, Philippines, and older countries like Thailand and Cambodia that continue to be areas of concern.*”

##### Investigative Methods

Participants highlighted difficulties in identifying victims (“*Many victims are not showing red flags.*”) and how to extricate relevant information from databases (“*Categorisation of database information does not generate new knowledge.*”). There was also a call to collaborate with other counties in on-going or future investigations:
“*One of the issues we raised was prevention. In the UK, we haven’t got a bottomless pit of resources of money. Let’s deal with our offenders going abroad and deal with them at source. They may be men, and they may be coming from here, but let’s deal with our own offenders first and see what works, and then spread that. Deal with the root cause.*”

##### Enhancing Knowledge about the Offending

Participants acknowledged that there is still limited knowledge about transnational offending pathways: Is it individual offenders or is it organised? Who do you pick up? How are groups accessed? One participant stated:
“*We were recognising that CSEM is all over, but why do people travel to do it? There are various hypotheses—high-risk sex offenders who carefully monitor might choose to do that. Respected individuals who might not do it in this country would go abroad—a process of normalisation might reinforce that. We were looking at could we understand the pathways by which people ended up doing that. Part of that would be what would come out of investigations leading to convictions, but part of it might come from interviewing people about how their pathways developed. It could be that it’s through sharing exploitative material, networking—building up to it from that. Or it might be that there’s knowledge of certain places to go to—how do people get that knowledge? From an intervention viewpoint, those processes might have points that could be disrupted.*”

##### Maintaining a Victim-Centric Stance

Participants wondered how they can implement a victim-centric policing model when managing cases of transnational offenders, especially when providing for the needs of victims in, or from other jurisdictions:
“*It is very difficult when you’ve got different levels of victim support and training—what exactly are we looking to prevent? Because a lot of these problems on the ground are not exactly preventable.*”

#### 3.1.3. Geographical Issues

Participants highlighted the need to identify and share knowledge on geographical areas of concern, as well as considering event-specific offending (such as surrounding the World Cup). Key markers of target areas were poverty and corruption:
“*We were saying corruption is a real problem—it leads on from poverty. The reverse is: the offenders are rich, they can buy people off. They can buy the family off, they can groom the family. They can pay police officers off. Families will take any amount of money for a child. So poverty, corruption and richness together is the perfect triangle of abuse, isn’t it?*”

Participants also highlighted how such markers could be used to identify areas of future concern.

A second issue that arose here was how Western culture might be perceived abroad:
“*Maybe it’s not just people travelling abroad … maybe they are more sexualised here? Is this a cultural phenomenon? Is [TCSA] actually a specific issue or is it just sexual abuse in general happening in different fora with new technology?*”“*We had a presentation at a hotel recently, there was a person from serious crime office talking about CSE [child sexual exploitation]—1.3 million children were abused in this country [UK]. (…) Why don’t we talk about this country? We have 10,000 refugee children missing in Europe. It’s much easier to talk about what’s happening in Vietnam—it makes us feel more comfortable, but that’s the way it is.*”

Finally, participants were considering the “borderless role” of technology and the dark web:
“*It depends on the level of sophistication of the offender you see. The newbies will go in social media—which are controlled by the police. We’re talking now about the dark social (not necessarily dark web), but those that can’t be encrypted such as Snapchat, where there’s no potential of looking. So if I’m an offender and I start a network with others who bring other people, then I can organise my own network.*”

#### 3.1.4. Tourism/Hospitality Issues

Interestingly, issues surrounding the tourism and hospitality sector emerged as a separate sub-theme. Participants explored the role of hotels (mainly, hotel receptions), and how they can play a part in the prevention of TCSA, and highlighted initiatives such as a “*Hotel Watch*” (e.g., https://sussex.police.uk/advice/protect-your-business/hotel-watch/). Participants also reflected on novel tourism developments, such as *AirBnB*, which emerged as both a “digital disruptor” to how safety, accountability, and regulation operates within the industry, but also as a new challenge to understanding how people use informal space via transactions [58]. 

### 3.2. Priority Ranking

Participants were asked to record any additional factors for consideration on post-it notes. These highlighted nine separate research questions that could be summarised into three key themes: Offending Behaviour; Victim Safeguarding; and Culturally-Sensitive Responding. Participants were asked to rate these in terms of their priority, highlighting the need for systematic information gathering and victim safeguarding as key priorities (see Table 1).

## 4. Discussion

This study investigated the issues related to TCSA from a multi-agency and multi-professional stakeholder perspective. Data were collected during a roundtable discussion event in London with stakeholder representatives from academia, law enforcement, tourist industry and non-profit child protection sectors. Participants discussed issues related to the key challenges of TCSA, the affected countries, response strategies, and knowledge gaps. The breadth of responses from attendees underscores the scale of the TCSA problem, and the challenges in responding to it. The discussions highlight four key themes as future priority areas (see Figure 1). 

### 4.1. Key Theme 1: Offender Typologies

A recurring theme from attendees’ discussions was the extent to which the field is encountering a new type of offender, or a new pathway to child sexual abuse behaviour. Key information required here is the analysis of features of geographical target areas as well as the role of technology and Internet access in target countries, to inform preventative strategies. A sub-question that emerged was the appropriate selection and analysis of information collected in relevant databases. Overall, participants acknowledged that the vast majority of known TCSA offenders are male (see also [4]) but were less clear on their psychological characteristics and how the pathway to TCSA was initiated (“*there’s a burning question about motivation there*”). Participants discussed their experiences and professional knowledge about the motivations of TCSA offenders and connected this to potential pathways to TCSA offending behaviour, with the view to identifying gateways for interrupting it and to further explore the role of the *dark web* in facilitating TCSA. Given the role of the Internet in facilitating TCSA, participants provided nuanced discussions related to the motivational continuum of (potential) offenders, and the overall function that the Internet has: “*How do people* [offenders] *get that knowledge* [of TCSA destinations]*? From an intervention viewpoint*, *those processes might have points that could be disrupted.*”

### 4.2. Key Theme 2: Victim-Centric Investigative Practice

In keeping with best practice procedures of investigative practice advocated by many stakeholders such as ECPAT [59] and the Declaration of Rome [60], attendees recognised the need to hold to account the multi-agency partners responsible for preventing re-victimisation (e.g., during investigation proceedings) and the provision of support services provided for the victims (and their families). Participants highlighted the need for good investigative practice to be shared across jurisdictions, and for greater clarity in terms of accountability and responsibility. Indeed, there was also the call to develop a shared definition of what constitutes effective support for victims, and for it to be then embedded in a culturally responsive framework. This connects with the victim-centred approach that has come to be prioritised across multi-agency responses to all forms of sexual offending. However, there was also recognition that this may not always be possible across jurisdictions where accountability of victim follow-up and support is unavailable. 

### 4.3. Key Theme 3: Prevalence and Definitions

There was a detailed discussion regarding the “hotspot countries” of TCSA, and the need to identify key markers of such countries to be able to foresee future vulnerable countries. Whilst law enforcement can locate regions with high levels of recorded TCSA, that is not to rule out other potential hotspot regions that are as yet unknown. Connected with the offender typologies theme, further data is required to critically evaluate where certain offender-types travel, why, and in what numbers. Other definitional issues arose with broad terms such as *culture* or *poverty*, and a suggestion to develop a shared language that does not overemphasise cultural differences. A final definitional and conceptual issue was the distinction of *victim* vs. *perpetrator* in light of the challenges from self-generated illegal content [45]. Whilst this analysis clearly argues for a victim-centric approach, there is much complexity underlying how we approach the concept of victimhood and perpetration within TCSA when there are significant mediating factors at play (*“… a lot of these problems on the ground are not exactly preventable”*).

### 4.4. Key Theme 4: Collaboration

It was evident that collaborative work from all stakeholders was seen as the route to the most effective responses. This necessitates strategically targeted information-sharing to be able to cultivate a feasible strategy, a level of collaboration between partners abroad, and the establishment and strengthening of such links. Participants challenged the assumptions made about where, how and why TCSA occurs, with respect to the social, economic, political, and cultural considerations [39]. Equally, there was acknowledgement of the work of agencies such as ECPAT [51] and parts of the tourist industry [52] in establishing ethical policies against TCSA. However, there was also acknowledgement of the underestimation of TCSA and its likely global increase [4]. Consequently, there were interesting discussions between participants regarding the design of the most appropriate ethical framework when proposing to intervene at any level abroad. A separate but related issue concerned how best to intervene given limited resources; for example, there is a need for a more coordinated international response to address ‘at risk’ offenders journeying into the pathways to TCSA. This final theme connects all other themes by way of establishing a solution-focused response to the issues raised—namely, collaborative approaches to prevention and interruption of the preparatory stages to (re-)offending.

Finally, perhaps one surprising outcome of this study was the lack of any detailed discussion regarding the specific role of the technology industry and government in moderating content, detecting and eradicating TCSA networks, and collaborating with stakeholders. This is perhaps partly attributable to the absence of any technology industry or government representation at the roundtable event. Nonetheless, it is somewhat surprising that specific technological interventions were not explored given the frequency at which cyberspace was implicated in TCSA. Enhancing the role of this industry in the challenge to eradicate TCSA is regarded as a fundamental component within the literature [61,62]. 

### 4.5. Limitations

This study comprised of a self-selected sample of participants that was not fully representative of all stakeholders. Notably, there was no representation of victims or perpetrators, nor was there any technology industry or political representation. Further limitations were the small number of participants, and any potential biases arising from their different ideological views, differential power, and any related consequences of these differences. In addition, while this type of data collection generated a wide-ranging set of data, their variability affects the overall strength and generalisability of the data collected. However, this approach was justified given the exploratory nature of this study, with the aim for future research to investigate each issue in-depth and develop clear action plans. 

## 5. Conclusions

The findings in this study lend support to further exploring how stakeholder collaboration could assist primary, secondary, and tertiary prevention approaches [63] to TCSA. Specifically, the participants’ prioritisation of the research questions identified throughout the day offers a potential starting point for future collaborations. It followed that offending behaviour (how offenders know where to go) and victim safeguarding (ensuring support and accountability for responses to victims) should be seen as the primary areas for multi-agency collaboration. 

To that end, this roundtable discussion highlighted the scope and motivation for collaboration between academia, front-line operations and charities. This research emphasised the need for investment in further resources that can address the knowledge gaps that are currently limiting the ability of stakeholders to tackle TCSA. For example, the sharing of data between academia, industry, charities, and law enforcement across jurisdictions can enhance knowledge regarding the psychology of the continuum of TCSA offender motivations and the respective offending behaviours, in both online and offline environments. In the next steps, the key themes identified need to be translated into clear action plans to be addressed in work going forward. 

## Figures and Tables

**Figure 1 ijerph-16-00243-f001:**
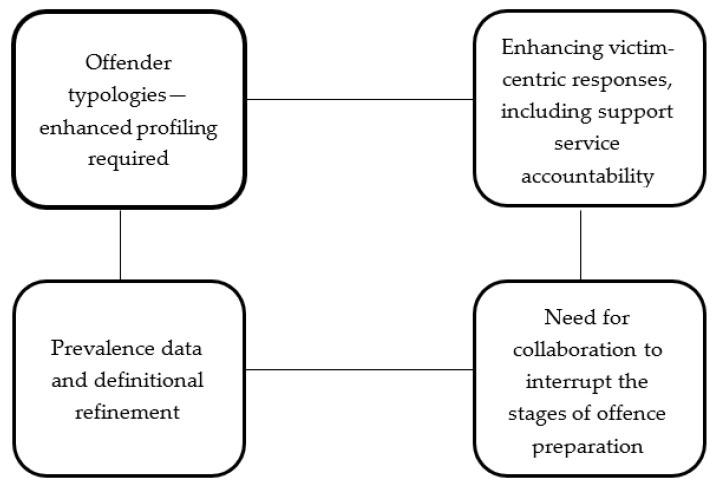
Thematic summary of overall priorities identified at the roundtable discussion.

**Table 1 ijerph-16-00243-t001:** Identified research questions, prioritised by the group (frequency rating).

Theme	Research Question	Priority Ranking
Offending Behaviour	How do offenders know to go to certain locations/distribution channels?	7
Victim Safeguarding	What support is available to children/victims in affected countries (e.g., safe housing, psychological support, compensation)?	6
Victim Safeguarding	In cases of compensation, how can we ensure there is no meta-abuse of the victim, and how can policy ensure such accountability?	5
Victim Safeguarding	In some countries, victims of child sexual abuse may be seen as criminally liable (differences in legislation between countries); issue of victim blaming/stigma surrounding victimisation, and power	5
Offending Behaviour	What types of transnational child sexual abuse exists?	5
Offending Behaviour	What are the profiles of perpetrators in-country vs. foreign visitors (travelling sex offenders)? How do live-streaming offences fit in?	5
Culturally-Sensitive Responding	Western way of thinking in a non-Western problem—how can we think/act globally?	4
Offending Behaviour	What is the impact of online child sexual abuse, networking, and travelling on contact sex offending?	4
Victim Safeguarding	Can you do research with the child victim? What are the consequences to the child? What are their perceptions of the help provided?	3

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
