# Peer review of "Transnational Child Sexual Abuse: Outcomes from a Roundtable Discussion"

_ijerph, 2019, doi:10.3390/ijerph16020243_

Round 1

Reviewer 1 Report

My comments and suggestions are attached.

Author Response

Many thanks for these points. As evidenced in the track-changes, we have corrected all suggestions, excluding point 9 where we feel the provided reference justifies the inclusion of ‘Orchid Club’. We hope this can inform the readers further.

Please see the attached file for a detailed response. 

Reviewer 2 Report

General: Interesting topic, well presented. Flow of language: acceptable

Title: suitable

Text structure and content: both are adequate

The reference list: most authors with important research in this area  are included.

Author Response

Thank you for these points. We hope that the changes prompted by the other reviewers will add to the flow of language, in addition to further strengthening the overall text structure. 

Please see the attached letter for a detailed response.

Reviewer 3 Report

Thank you for the opportunity to read this paper. In general you report on an interesting issue that requires ongoing collaborative research and action. My major comments relate to the rigor of the reporting of your research methodology. I would suggest an expansion of the discussion about the research process - consider including a more explicit link between your aims, your method and your outcome as well as more detail about who was there, who was not, how did you manage different ideological views, differential power and ethical issues. Need more about the stakeholders in order to validate their contribution in this issue. Finally presentation can be improved with a thorough edit : line 18 - use of acronym prior to full wording line 30,31, 33 - capitalization after semi colon and colon is unnecessary You interchangeably use individuals, men and offender to refer to the same group - I suggest you get consistent and explain your choice. Line 50, 54, 78, 93/94, 159/160 you make inappropriate use of a semi colon - it is not for random lists. The inconsistent presentation of your data distracts from your point and makes it difficult for a reader to follow the development of your conclusions - you varyingly use indented quotes, italics in brackets, italics in a sentence and quotation marks non italicized sentences.

Author Response

Thank you for this feedback. We agree that there may need to be ‘more about the stakeholders in order to validate their contribution’. However, we are limited to expand due to the agreed confidentiality of the participants and their organisational affiliation. Thus we have added this point to our limitations subsection.

We have edited these lines as suggested.

We have chosen to use ‘offender’ to most appropriately reflect their actions, and indeed the legislative terminology (note where other labels are used in quotes, no edits were made).

These lines have now been remedied.

We have now indented long-form quotes with consistent use of italics. A small number of short-form quotes (one sentence length) remain non-indented because we feel this adds to the discursiveness of respective point being outlined. We feel that these valuable data presentation points have now been addressed to sufficiently improve the readability.

Please see the attached letter for a detailed response.
